# Identifying good directions to escape the NTK regime and efficiently learn low-degree plus sparse polynomials

**Eshaan Nichani**
Princeton University
eshnich@princeton.edu

**Yu Bai**
Salesforce Research
yu.bai@salesforce.com

**Jason D. Lee**
Princeton University
jasonlee@princeton.edu

## Abstract

A recent goal in the theory of deep learning is to identify how neural networks can escape the "lazy training," or Neural Tangent Kernel (NTK) regime, where the network is coupled with its first order Taylor expansion at initialization. While the NTK is minimax optimal for learning dense polynomials [25], it cannot learn features, and hence has poor sample complexity for learning many classes of functions including sparse polynomials. Recent works have thus aimed to identify settings where gradient based algorithms provably generalize better than the NTK. One such example is the "QuadNTK" approach of Bai and Lee [7], which analyzes the second-order term in the Taylor expansion. Bai and Lee [7] show that the second-order term can learn sparse polynomials efficiently; however, it sacrifices the ability to learn general dense polynomials.

In this paper, we analyze how gradient descent on a two-layer neural network can escape the NTK regime by utilizing a spectral characterization of the NTK [39] and building on the QuadNTK approach. We first expand upon the spectral analysis to identify "good" directions in parameter space in which we can move without harming generalization. Next, we show that a wide two-layer neural network can jointly use the NTK and QuadNTK to fit target functions consisting of a dense low-degree term and a sparse high-degree term – something neither the NTK nor the QuadNTK can do on their own. Finally, we construct a regularizer which encourages the parameter vector to move in the "good" directions, and show that gradient descent on the regularized loss will converge to a global minimizer, which also has low test error. This yields an end to end convergence and generalization guarantee with provable sample complexity improvement over both the NTK and QuadNTK on their own.

## 1 Introduction

In recent years, deep learning has acheived a number of practical successes, in domains spanning computer vision, natural language processing, reinforcement learning, and the sciences. Despite these impressive empirical results, the theory underlying deep learning is far from complete. In fact, the dual questions of optimization – the mechanism by which neural networks trained with gradient descent are able to interpolate training data despite the nonconvexity of the loss landscape – and

36th Conference on Neural Information Processing Systems (NeurIPS 2022).

generalization – why these solutions found by gradient descent require relatively few samples to generalize – are still not well understood.

One successful approach for understanding optimization has been the Neural Tangent Kernel (NTK) theory [41, 29, 15, 20]. The NTK approach couples the gradient descent dynamics of a wide neural network under a specific initialization to the gradient descent dynamics of a particular kernel regression problem, with a random, initialization dependent kernel. In the limit of infinite width, this kernel converges almost surely to a deterministic kernel, also referred to as the NTK, and properties of this kernel and its corresponding Reproducing Kernel Hilbert Space can be studied.

However, recent work has shown that the NTK theory fails to explain the generalization capabilities of neural networks. While the equivalence between neural networks in the NTK regime and kernel methods implies that such models perform no better than kernels, in practice, neural networks have been shown to outperform kernel methods on a number of tasks [5, 33]. Theoretically, a recent line of work [25, 39, 38] has provided a precise statistical analysis of the generalization properties of rotationally invariant kernels on the unit sphere, which includes the NTK. [25] proves a sample complexity lower bound for such kernels, showing that $d^k$ samples are needed to learn any degree $k$ polynomial in $d$ dimensions. As a result, the NTK is no better than a polynomial kernel, and cannot adapt to low-dimensional structure.

The limitations of the NTK can be further understood from the linearization perspective. Consider a two-layer neural network $f(\mathbf{x}; \mathbf{W})$ with input $\mathbf{x}$, width $m$, first layer weights initialized as $\mathbf{W}_0 \in \mathbb{R}^{d \times m}$, second layer weights $\mathbf{a} \in \mathbb{R}^m$, activation function $\sigma$, and displacement from initialization $\mathbf{W} \in \mathbb{R}^{d \times m}$:

$$f(\mathbf{x}; \mathbf{W}) = \frac{1}{\sqrt{m}} \sum_{r=1}^{m} a_r \sigma(\mathbf{w}_{0,r}^T \mathbf{x} + \mathbf{w}_r^T \mathbf{x}). \tag{1}$$

For simplicity we assume the second layer weights are held fixed, and so $\mathbf{W}$ are the trainable parameters. The NTK theory states that when $\mathbf{W}$ has small norm, the gradient descent dynamics can be well approximated by replacing the model with its first-order Taylor expansion about the initialization:

$$f(\mathbf{x}; \mathbf{W}) \approx f(\mathbf{x}; \mathbf{0}) + \frac{1}{\sqrt{m}} \sum_{r=1}^{m} a_r \sigma'(\mathbf{w}_{0,r}^T \mathbf{x}) \mathbf{x}^T \mathbf{w}_r = f(\mathbf{x}; \mathbf{0}) + \text{vec}(\mathbf{W})^T \varphi(\mathbf{x}). \tag{2}$$

Here, $\{\mathbf{w}_r\}_{r \in [m]}$ and $\{\mathbf{w}_{0,r}\}_{r \in [m]}$ are the columns of $\mathbf{W}$ and $\mathbf{W}_0$ respectively, $\varphi(\mathbf{x}) := \text{vec}(\nabla_\mathbf{W} f(\mathbf{x}; \mathbf{W})|_{\mathbf{W}=\mathbf{0}})$ is a random feature vector with norm independent of $m$, and

$$f_L(\mathbf{x}; \mathbf{W}) := \text{vec}(\mathbf{W})^T \varphi(\mathbf{x}) \tag{3}$$

is hereafter referred to as the *linear term*. Ignoring $d$ dependence, there exists a global minimizer with $\|\mathbf{W}\|_F \simeq 1$ and $\|\mathbf{w}_r\|_2 \simeq m^{-1/2}$, and thus due to local convexity gradient descent will stay in this small norm ball around the initialization while interpolating the training data. This small movement of each individual neuron gives rise to the name *lazy training* for networks in the NTK regime [15]. The equivalence to kernel methods and the poor generalization of neural networks in NTK regime, along with their failure to describe the dynamics of neural networks in practice, motivate our goal to understand how neural networks can escape the NTK regime. We concretely ask the following question:

**Q: How can we encourage each neuron to move $\gg m^{-1/2}$, thus escaping the NTK regime? And does this allow us to break the NTK sample complexity lower bounds?**

### 1.1 Motivation

**Escaping the NTK Regime.** To answer this, we invoke the statistical characterization of the NTK developed in [25, 39, 38] to understand the mechanism by which it overfits to the training data. For $d^k \ll n \ll d^{k+1}$, consider a dataset of $n$ training samples $(\mathbf{x}_i, y_i)$, where the $\mathbf{x}_i$ are sampled i.i.d from $\mathcal{S}^{d-1}(\sqrt{d})$ (the $d$-dimensional sphere of radius $\sqrt{d}$) and $y_i = f^*(\mathbf{x})$ for an unknown function $f^*$. Montanari and Zhong [39] decompose the SVD of the empirical feature matrix $\mathbf{\Phi} \in \mathbb{R}^{n \times md}$ into the block matrix form

$$\mathbf{\Phi} = \begin{bmatrix} \varphi(\mathbf{x}_1)^T \\ \cdots \\ \varphi(\mathbf{x}_n)^T \end{bmatrix} = \begin{bmatrix} \mathbf{U}_1 & \mathbf{U}_2 \end{bmatrix} \begin{bmatrix} \tilde{\mathbf{\Lambda}}_1 & \mathbf{0} \\ \mathbf{0} & \tilde{\mathbf{\Lambda}}_2 \end{bmatrix} \begin{bmatrix} \mathbf{V}_1^T \\ \mathbf{V}_2^T \end{bmatrix}, \tag{4}$$

where $\mathbf{U}_1, \tilde{\boldsymbol{\Lambda}}_1, \mathbf{V}_1$ are the top $r$ singular values/vectors and $\mathbf{U}_2, \tilde{\boldsymbol{\Lambda}}_2, \mathbf{V}_2$ are the bottom $n - r$. Here, $r = O(d^k)$ is chosen specifically so that $\mathbf{V}_1$ is the "high-variance" subspace which can express polynomials of degree $\leq k$ and generalizes well [9], while $\mathbf{V}_2$ is used to interpolate the training data while not affecting generalization.

[25, 39] show that the NTK will learn the projection of $f^*$ onto degree $k$ polynomials. Furthermore, [9, 26] show that gradient descent will first move in the subspace spanned by $\mathbf{V}_1$ to learn this projection, then move in the $\mathbf{V}_2$ directions to interpolate the training data, while not affecting the test predictions. This first stage is still desirable for our goal – if the network can fit and generalize from part of the signal using the NTK, then it should, and previous work [27] has shown that for general networks gradient descent will learn the optimal degree-1 polynomial in the early stages of training. The $\mathbf{V}_2$ directions, however, are "bad" directions the parameters should avoid moving in, as they are only used for the NTK to overfit. Instead, $\mathbf{W}$ should move in the null space of $\boldsymbol{\Phi}$, where $\|\mathbf{W}\|_F$ can be $\Omega(1)$ while keeping the evaluation of $f_L(\cdot; \mathbf{W})$ on the training data bounded by $O(1)$. This heuristic argument yields our first criterion for escaping the NTK regime.

**Goal #1: Move minimally in the $\mathbf{V}_2$ directions.**

**Generalizing to Test Data.** While the previous criterion prevents the network from overfitting with the NTK, we must also prevent a movement of $\|\mathbf{W}\|_F \gg 1$ from causing the test predictions to explode. Since $\|\varphi(\mathbf{x})\|_2 = \Theta_m(1)$, it is a priori possible for $f_L(\mathbf{x}; \mathbf{W}) = \text{vec}(\mathbf{W})^T \varphi(\mathbf{x})$ to be $\gg 1$ on the population, which would necessarily cause a large test loss.

To identify a set of good directions, define the *feature covariance matrix* $\boldsymbol{\Sigma} \in \mathbb{R}^{md \times md}$ by

$$\boldsymbol{\Sigma} = \mathbb{E}_{\mathbf{x} \sim \mathcal{S}^{d-1}(\sqrt{d})} \left[ \varphi(\mathbf{x}) \varphi(\mathbf{x})^T \right]. \tag{5}$$

Our first technical contribution is a characterization of the eigendecomposition of $\boldsymbol{\Sigma}$. Let the top $r$ eigenvectors of $\boldsymbol{\Sigma}$ be $\mathbf{Q}_1$, the next $s$ eigenvectors be $\mathbf{Q}_2$, and the bottom $md - r - s$ eigenvectors be $\mathbf{Q}_3$, where $\boldsymbol{\Lambda}_1, \boldsymbol{\Lambda}_2, \boldsymbol{\Lambda}_3$ are the corresponding diagonal matrices of eigenvalues, and $r = \Theta(d^k), s = d^{\Theta(k)}$. We use the results of [39, 38] to show that $\boldsymbol{\Sigma}$ has an eigenvalue gap in that $\lambda_{min}(\boldsymbol{\Lambda}_1) \gg \lambda_{max}(\boldsymbol{\Lambda}_2)$ and $\lambda_{min}(\boldsymbol{\Lambda}_2) \gg \lambda_{max}(\boldsymbol{\Lambda}_3)$, and furthermore that $\mathbf{Q}_1$ can fit arbitrary degree $\leq k$ polynomials. This partitioning of the eigenvectors tells us that $\mathbf{Q}_1$ are informative, large eigenvalue directions which help the NTK to learn a low degree signal, $\mathbf{Q}_2$ are the medium directions which will cause the test predictions to grow too large if $\mathbf{W}$ moves too far from initialization, and $\mathbf{Q}_3$ are the "good" directions which $\mathbf{W}$ is free to move a distance of $\gg 1$ in. This yields the following criterion for generalizing well:

**Goal #2: Move in the $\mathbf{Q}_3$ directions, but minimally in the $\mathbf{Q}_2$ directions.**

One challenge is that we cannot distinguish the $\mathbf{Q}_2$ directions from the $\mathbf{Q}_3$ directions with $d^k$ samples. Nevertheless, the existence of an eigenvalue gap will allow us to constrain movement in $\mathbf{Q}_2$.

**The Quadratic NTK.** Once we have moved $\gg 1$ from the initialization, we can no longer couple to the network's linearization. The network is still, however, in a local regime, and we instead can couple the training dynamics to the second-order Taylor expansion of our model, where the second-order term is denoted the *Quadratic-NTK* [7] $f_Q(\mathbf{x}; \mathbf{W})$:

$$f(\mathbf{x}; \mathbf{W}) \approx f(\mathbf{x}; \mathbf{0}) + \text{vec}(\mathbf{W})^T \varphi(\mathbf{x}) + \frac{1}{2\sqrt{m}} \sum_{r=1}^{m} \sigma''(\mathbf{w}_{0,r}^T \mathbf{x})(\mathbf{x}^T \mathbf{w}_r)^2 \tag{6}$$

$$= f(\mathbf{x}; \mathbf{0}) + f_L(\mathbf{x}; \mathbf{W}) + f_Q(\mathbf{x}; \mathbf{W}). \tag{7}$$

Bai and Lee [7] showed that $f_Q(\mathbf{x}; \mathbf{W})$ can effectively learn low-rank polynomials with better sample complexity than the NTK. In particular, they show that $d^k$ samples are needed to fit a target function of the form $f^*(\mathbf{x}) = (\beta^T \mathbf{x})^{k+1}$, an improvement over the $d^{k+1}$ samples needed by the NTK. In doing so, however, the QuadNTK sacrifices its ability to learn general dense polynomials; furthermore, Bai and Lee [7] require a randomization trick to artificially delete the $f_L$ term. A later followup work [8] showed that in a number of standard experimental settings, the second-order Taylor expansion of the network better tracks the true gradient descent dynamics and acheives lower test loss than the network's linearization (i.e NTK) does. However, there is no existing result which shows that both the linear term and the quadratic term can provably learn a component of the signal.

Based on the preceeding motivation, we thus aim to show that we can jointly utilize both the NTK and the QuadNTK to learn a larger class of functions than either the NTK or QuadNTK can learn on their own.

## 1.2 Our Contributions

With the previous intuition in hand, we outline the main contributions of our work. We first prove a technical result on the eigendecomposition of $\Sigma$, and show that the eigenvectors can indeed be partitioned into 3 categories corresponding to large (degree $\leq k$), medium ("bad") and small ("good") eigenvalues. We then construct a regularizer, depending only on the covariate distribution and initialization, that enforces goals 1 and 2 by preventing the parameters from moving in either of the bad sets of directions ($\mathbf{V}_2$ and $\mathbf{Q}_2$). Furthermore, we show how to jointly use the NTK and QuadNTK to fit a target signal $f^*$ consisting of an arbitrary degree $\leq k$ component and sparse degree $k+1$ component. The key technical challenge is to construct a solution with large enough movement so that the QuadNTK can fit the high degree term and hence improve generalization, while simultaneously preventing this large movement from interfering with the NTK training predictions (Goal 1) or greatly increasing test loss (Goal 2). Our main result, Theorem 1, is that gradient descent on a polynomially wide two-layer neural network converges to an approximate global minimizer of the regularized loss function, which generalizes well to the test distribution. As a result, we show $d^k$ samples are needed to learn $f^*$ up to vanishingly small test loss. This ultimately gives us the "best of both worlds", as we leverage both the linear and quadratic term to learn the target $f^*$ with sample complexity better than either the NTK or QuadNTK alone. Overall, our work identifies which directions weights can move further from initialization in and provably generalize better than the NTK.

The outline of our paper is as follows. In Section 2 we formally define the problem setup. In Section 3 we define our regularizers, and present Theorem 1. Section 4 is an outline of the proof of Theorem 1, which we split into four components – expressing $f^*$ with the linear-plus-quad model, showing the optimization landscape has favorable geometry, a gradient descent convergence result, and a generalization bound. We conclude with experiments supporting our main theorem and demonstrating the relevance of the low-degree plus sparse task to standard neural networks.

## 1.3 Related Work

The NTK approach [41, 29, 15, 32, 20], which couples a neural network to its linearization at initialization, has been utilized to show global convergence of gradient descent on neural networks [19, 35, 47, 4]. The equivalence to kernel methods has also been used to prove generalization bounds, based on generalization bounds for kernels [6, 11, 3]. However, neural networks have been shown to perform far better than their NTK in practice [5, 33]. Further, [25] shows that kernels cannot adapt to low-dimensional structure, and proves a sample complexity lower bound of $d^k$ samples needed to learn a degree $k$ polynomial in $d$ dimensions. A number of recent works [44, 18, 46, 1, 23, 2, 24, 16, 45, 36, 14, 37] have thus aimed to provide examples of learning problems where a neural network trained with a gradient-based algorithm has a provable sample complexity improvement over any kernel method.

One such approach has been to understand higher-order approximations of the training dynamics [7, 8, 12, 28]. Here, the network is no longer coupled to its linearization, but rather higher order terms in the Taylor expansion. On the empirical side, [8] shows that these higher order Taylor expansions better track the optimization dynamics and can obtain lower test loss. Theoretically, [7, 12] prove that the second-order term, the QuadNTK, can be used to obtain sample complexity improvements. However, the QuadNTK has poor sample complexity for learning dense polynomials, and [7, 12] do not consider training on the original network, but rather only the second-order term after the linear term has been deleted. This paper, on the other hand, provides an end-to-end convergence and generalization result for training on the full two-layer neural network. We leverage the NTK to efficiently learn polynomials with both a dense and sparse component, and are thus the first work showing that both the linear and quadratic term can learn part of the signal.

The technical results in our paper rely on the statistical characterization of the NTK developed in the series of works [25, 39, 38]. Furthermore, our optimization results rely on a line of work showing that quadratically parameterized models have nice landscape properties such as all second-order saddle points are global minima [21, 22, 41, 18]; the fact that gradient descent avoids saddle points [21, 34, 30, 31] can then be used to show convergence.

## 2 Preliminaries

### 2.1 Problem Setup

Our problem setup is the standard supervised learning setting. Our dataset $\mathcal{D}_n = \{(\mathbf{x}_i, y_i)\}_{i \in [n]}$, has $n$ samples, where $(\mathbf{x}_i, y_i) \in \mathcal{X} \times \mathcal{Y}$ are sampled i.i.d from a distribution $\mu$ on $\mathcal{X} \times \mathcal{Y}$. $\mu$ is defined so that $(\mathbf{x}, y) \sim \mu$ satisfies $x \sim \text{Unif}(\mathcal{S}^{d-1}(\sqrt{d}))$, the uniform distribution on the $d$-dimensional sphere of radius $\sqrt{d}$, and $y = f^*(\mathbf{x})$ for some deterministic, unknown function $f^* : \mathcal{S}^{d-1}(\sqrt{d}) \to \mathbb{R}$.

We assume that $d^k \ll n \ll d^{k+1}$ for some integer $k$, and that the target $f^*$ has the following low-degree plus sparse structure:

**Assumption 1** (Low-degree plus sparse signal). *Let $f^*(\mathbf{x}) = f_k(\mathbf{x}) + f_{sp}(\mathbf{x})$, where*

- *$f_k(\mathbf{x})$ is an arbitrary degree $\leq k$ polynomial with $\mathbb{E}_{\mathbf{x} \sim \mu}[f_k(\mathbf{x})^2] = 1$. (Low Degree)*

- *$f_{sp}(\mathbf{x}) = \sum_{i=1}^{R} \alpha_i (\beta_i^T \mathbf{x})^{k+1}$ where $|\alpha_i| \leq 1, \|\beta_i\|_2 = 1$. (Sparse)*

We aim to fit $f^*$ with $f(\mathbf{x}; \mathbf{W})$, a two-layer neural network as defined in (1). Here, $\mathbf{W} = [\mathbf{w}_1, \dots, \mathbf{w}_r] \in \mathbb{R}^{d \times m}$ is the first layer weight's distance from initialization and is the trainable parameter. $\mathbf{W}_0 = [\mathbf{w}_{0,1}, \dots, \mathbf{w}_{0,r}]$ denotes the first layer weight at initialization, and and $\mathbf{a} = [a_1, \dots, a_m]^T \in \mathbb{R}^m$ is the second layer weight, which is held fixed throughout training.

We consider the following *symmetric initialization* of $\mathbf{a}, \mathbf{W}_0$, which ensures that $f(\cdot; \mathbf{0}) = 0$ identically.

$$a_1 = \dots = a_{m/2} = 1 \qquad a_{m/2+1} = \dots = a_m = -1 \qquad (8)$$

$$\{\mathbf{w}_{0,r}\}_{r \leq m/2} \sim_{i.i.d} \mathcal{S}^{(d-1)}(1) \qquad \mathbf{w}_{0,m/2+r} = \mathbf{w}_{0,r} \qquad (9)$$

$\sigma \in C^2(\mathbb{R})$ is our nonlinear activation function. We make the following assumption on $\sigma$:

**Assumption 2.** *The activation $\sigma$ satisfies $\|\sigma\|_\infty, \|\sigma'\|_\infty, \|\sigma''\|_\infty < 1$.*

We also require $\sigma', \sigma''$ to satisfy Assumption 3, a particular technical condition on their harmonic expansions. These two assumptions are satisfied by commonly used activations, such as the sigmoid with generic shift $b$: $\sigma(z) = \frac{1}{1+\exp(b-z)}$ .

We assume the loss function $\ell : \mathbb{R} \times \mathbb{R} \to \mathbb{R}^{\geq 0}$ satisfies $\ell(y, z) \leq 1$, $\ell(y, y) = 0$, $\ell(y, z)$ convex in $z$, and $\|\frac{\partial}{\partial z}\ell\|_\infty, \|\frac{\partial^2}{\partial z^2}\ell\|_\infty, \|\frac{\partial^3}{\partial z^3}\ell\|_\infty \leq 1$. The empirical loss $\hat{L}$ and population loss $L$ are defined as

$$\hat{L}(\mathbf{W}) = \mathbb{E}_n\left[\ell(y, f(\mathbf{x}; \mathbf{W}))\right] \qquad L(\mathbf{W}) = \mathbb{E}_\mu\left[\ell(y, f(\mathbf{x}; \mathbf{W}))\right], \qquad (10)$$

where for a function $g(\mathbf{x}, y)$, $\mathbb{E}_n[g(\mathbf{x}, y)] := \frac{1}{n}\sum_{i=1}^{n} g(\mathbf{x}_i, y_i)$ denotes the empirical expectation, while $\mathbb{E}_\mu[g(\mathbf{x}, y)]$ denotes the population expectation over $(\mathbf{x}, y) \sim \mu$.

**Notation.** For $f \in L^2(\mathcal{S}^d(\sqrt{d}), \mu)$, define $\|f\|_{L^2} := \|f\|_{L^2(\mathcal{S}^d(\sqrt{d}),\mu)} = \left(\mathbb{E}_{x \sim \mu}[(f(\mathbf{x}))^2]\right)^{1/2}$. We use big $O$ notation to ignore absolute constants that do not depend on $n, d, m$, as well as polynomial dependencies on the rank $R$. We write $a_d \lesssim b_d$ if $a_d = O(b_d)$, $a_d \ll b_d$ if $\lim_{d \to \infty} a_d/b_d = 0$. We also use $\tilde{O}$ notation to ignore terms that depend logarithmically on $d$. We also treat $k = O(1)$. Finally, all our results hold for $d > C$, where $C$ is a universal constant. For a matrix $\mathbf{A}$, we let $\|\mathbf{A}\|_F$ be its Frobenius norm, $\|\mathbf{A}\| = \|\mathbf{A}\|_{op}$ be the operator norm, and $\|\mathbf{A}\|_{2,p} := (\sum_i \|\mathbf{a}_i\|_2^p)^{1/p}$ be the $2, p$ norm.

### 2.2 Linear and Quadratic Expansion

For $\mathbf{W}$ small, $f(\cdot; \mathbf{W})$ can be approximated by its second order Taylor expansion about $\mathbf{W}_0$:

$$f(\mathbf{x}; \mathbf{W}) \approx \frac{1}{\sqrt{m}} \sum_{r=1}^{m} a_r \sigma'(\mathbf{w}_{0,r}^T \mathbf{x})\mathbf{x}^T\mathbf{w}_r + \frac{1}{2}a_r\sigma''(\mathbf{w}_{0,r}^T\mathbf{x})(\mathbf{x}^T\mathbf{w}_r)^2. \qquad (11)$$

We define $f_L(\mathbf{x}; \mathbf{W})$, $f_Q(\mathbf{x}; \mathbf{W})$ to be the linear and quadratic terms of the network:

$$f_L(\mathbf{x}; \mathbf{W}) = \frac{1}{\sqrt{m}} \sum_{r=1}^{m} a_r \sigma'(\mathbf{w}_{0,r}^T \mathbf{x}) \mathbf{x}^T \mathbf{w}_r, \qquad f_Q(\mathbf{x}; \mathbf{W}) = \frac{1}{\sqrt{m}} \sum_{r=1}^{m} \frac{1}{2} a_r \sigma''(\mathbf{w}_{0,r}^T \mathbf{x})(\mathbf{x}^T \mathbf{w}_r)^2 \tag{12}$$

## 3 Main Theorem

Define the *NTK featurization map* $\varphi : \mathcal{S}^{d-1}(\sqrt{d}) \to \mathbb{R}^{md}$ as

$$\varphi := \text{vec}(\nabla_{\mathbf{W}} f(\mathbf{x}; \mathbf{W})|_{\mathbf{W}=\mathbf{0}}). \tag{13}$$

and the *feature covariance matrix* $\mathbf{\Sigma} \in \mathbb{R}^{md \times md}$ as

$$\mathbf{\Sigma} := \mathbb{E}_{\mathbf{x} \sim \mu} \left[ \varphi(\mathbf{x})\varphi(\mathbf{x})^T \right]. \tag{14}$$

Note that $\mathbf{\Sigma}$ depends only on the network at initialization and the input distribution, and not on the target function $f^*$. In practice, $\mathbf{\Sigma}$ can be approximated to arbitrary precision by using a large dataset of unlabeled data, or by computing the harmonic expansion of $\sigma'$, as detailed in Appendix A.

Let $\mathbf{\Sigma}$ admit the eigendecomposition $\mathbf{\Sigma} = \sum_{i=1}^{md} \lambda_i(\mathbf{\Sigma}) \mathbf{v}_i \mathbf{v}_i^T$, where the $\lambda_i(\mathbf{\Sigma})$ are nonnegative and nonincreasing. For $r \in [md]$, we let $\mathbf{\Pi}_{\leq r}$ be the projection operator onto $\text{span}(\mathbf{v}_1, \dots, \mathbf{v}_r)$, and let $\mathbf{\Pi}_{>r} = \mathbf{I}_{md} - \mathbf{\Pi}_{\leq r}$. Furthermore, define

$$\mathbf{\Sigma}_{\leq r} := \sum_{i=1}^{r} \lambda_i(\mathbf{\Sigma}) \mathbf{v}_i \mathbf{v}_i^T, \qquad \mathbf{\Sigma}_{>r} = \mathbf{\Sigma} - \mathbf{\Sigma}_{\geq r}. \tag{15}$$

We define our regularizers as follows

$$\mathcal{R}_1(\mathbf{W}; r) := \text{vec}(\mathbf{W})^T \mathbf{\Sigma}_{>r} \text{vec}(\mathbf{W}) \tag{16}$$

$$\mathcal{R}_2(\mathbf{W}; r) := \text{vec}(\mathbf{W})^T \mathbf{\Sigma}_{\leq r} \text{vec}(\mathbf{W}) \tag{17}$$

$$\mathcal{R}_3(\mathbf{W}; r) := \mathbb{E}_n \left[ (f_L(\mathbf{x}; \mathbf{\Pi}_{>r}\mathbf{W}))^2 \right] \tag{18}$$

$$\mathcal{R}_4(\mathbf{W}) := \|\mathbf{W}\|_{2,4}^8. \tag{19}$$

Intuitively, $\mathcal{R}_3$ constrains movement in the $\mathbf{V}_2$ directions to enforce Goal #1, $\mathcal{R}_1$ constrains movement in the $\mathbf{Q}_2$ directions to enforce Goal #2, and $\mathcal{R}_2, \mathcal{R}_4$ are weight-decay like terms necessary for generalization. Although $\mathcal{R}_1$ does not know the directions $\mathbf{Q}_2$, the eigenvalue gap between $\mathbf{Q}_2$ and $\mathbf{Q}_3$ ensures that whenever $\mathcal{R}_1$ is small, movement in $\mathbf{Q}_2$ must be small as well.

Given regularization parameters $\lambda = (\lambda_1, \lambda_2, \lambda_3, \lambda_4)$, define the regularized loss $L_\lambda(\mathbf{W})$ as

$$L_\lambda(\mathbf{W}) = \hat{L}(\mathbf{W}) + \lambda_1 \mathcal{R}_1(\mathbf{W}; r) + \lambda_2 \mathcal{R}_2(\mathbf{W}; r) + \lambda_3 \mathcal{R}_3(\mathbf{W}; r) + \lambda_4 \mathcal{R}_4(\mathbf{W}). \tag{20}$$

Finally, we train our model trained via perturbed gradient descent [31] with learning rate $\eta$ and noise level $\sigma^2$. That is, if $\mathbf{W}^t$ denotes the weights at time step $t$, the update is given by

$$\mathbf{W}^{t+1} = \mathbf{W}^t - \eta \left( \nabla_{\mathbf{W}} L_\lambda(\mathbf{W}^t) + \mathbf{\Xi}_t \right), \tag{21}$$

where $\mathbf{\Xi}_t \in \mathbb{R}^{d \times m}$ are i.i.d random matrices with each entry i.i.d $\mathcal{N}(\frac{\sigma^2}{md})$.

Given these definitions, we now present our main theorem:

**Theorem 1.** *Let $\varepsilon > 0$ be a target test accuracy, the number of samples be $n \gtrsim d^k \cdot poly(R) \cdot \max(\varepsilon^{-2}, \log d)$, and the width be $m = poly(n, d, R, \varepsilon^{-1})$. Let the sequence of iterates $\{\mathbf{W}^t\}_{t \geq 0}$ follow the update in (21) with initialization $\mathbf{W}^0 = \mathbf{0}$. Then, there exists a choice of parameters $(\lambda_1, \lambda_2, \lambda_3, \lambda_4, r, \sigma^2, \eta)$ such that with high probability over $\mathbf{W}_0, \mathcal{D}_n$, and $\{\mathbf{\Xi}_t\}_{t \geq 0}$, there exists a $\mathscr{T} = poly(m)$ such that the predictor $\hat{\mathbf{W}} := \mathbf{W}^{\mathscr{T}}$ satisfies $L(\hat{\mathbf{W}}) \leq \varepsilon$.*

**Remark 1.** Theorem 1 tells us $n = \tilde{\Theta}(d^k)$ samples are needed to learn the low-degree plus sparse function $f^*$. This is an improvement over the sample complexity needed to learn $f^*$ via the NTK, which is $\Omega(d^{k+1})$ samples as $f^*$ is a degree $(k+1)$-polynomial [25, 39]. This also improves over the upper bound for the sample complexity of the quadratic NTK given in [7], in which $\Omega(K^2 d^k)$ samples are needed to learn a rank $K$ polynomial of degree $k + 1$. This is polynomially worse than our bound since the dense, low-degree term $f_k$ can have rank $\gg d$.

# 4 Proof Sketch

The proof of Theorem 1 follows similar high-level steps to [7]. We first construct a $\mathbf{W}^* \in \mathbb{R}^{d \times m}$ which fits $f^*$ and is small on the regularizers. Next, we show that the optimization landscape of the regularized loss has a favorable geometry, and as a result that gradient descent converges to a global minimum. We conclude with a generalization bound to show that global minima have low test loss. Throughout the proof sketch, we emphasize how the regularizers encourage us to escape the NTK regime, and discuss the challenges posed by the existence of the $f_L$ term in the dynamics.

## 4.1 Expressivity

We begin by showing that the $f_L$ and $f_Q$ terms can fit the low degree and sparse components, respectively, of the signal. As in [25, 39], the derivations in this section rely on spherical harmonics; an overview of the technical results used are presented in Appendix A.

The following lemma shows that $f_Q(\mathbf{x}; \mathbf{W})$ can fit the sparse, high degree term in $f^*$.

**Lemma 1** (QuadNTK can fit high degree component). *Let $m \geq d^{2k}$. With high probability over the initialization, there exists $\mathbf{W}_Q$ such that*

$$\max_{i \in [n]} |f_Q(\mathbf{x}_i; \mathbf{W}_Q) - f_{sp}(\mathbf{x}_i)| \lesssim \frac{d^k}{\sqrt{m}} \quad and \quad \|\mathbf{W}_Q\|_{2,4}^4 \lesssim d^{k-1} \tag{22}$$

This generalizes the corresponding result in [7] to more activations. The proof of this lemma is presented in Appendix B.1.2.

Next, we show that $f_L(\mathbf{x}; \mathbf{W})$ can fit the low degree term. Here, we choose $r = n_k = \Theta(d^k)$, where $n_k$ is defined in Appendix A, to be the dimension of the subspace which can express degree $\leq k$ polynomials. We define $\mathbf{P}_{\leq k} = \mathbf{\Pi}_{\leq r}$ to be the projection on the top $n_k$ eigenvectors of $\mathbf{\Sigma}$.

**Lemma 2** (NTK can fit low degree component). *Let $m \geq d^{10k}$. With high probability, there exists $\mathbf{W}_L$ with $vec(\mathbf{W}_L) \in span(\mathbf{P}_{\leq k})$ such that*

$$\mathbb{E}_n[(f_L(\mathbf{x}; \mathbf{W}_L) - f_k(\mathbf{x}))^2] \lesssim \frac{d^k}{n} \quad and \quad \|\mathbf{W}_L\|_F^2 \lesssim d^{k-1} \tag{23}$$

The proof of this Lemma is presented in Appendix B.2.4, and relies on key lemmas from [39] relating the spherical harmonics of degree $\leq k$ to the eigenstructure of the kernel. A key intermediate result is Lemma 15, which characterizes the spectrum of the population covariance matrix $\mathbf{\Sigma}$. A similar result for random features was shown in [38]. Unlike [38], we do not characterize all the eigenvalues of $\mathbf{\Sigma}$; however, simply partitioning them into the three categories is sufficient for our purposes.

Finally, we use the $\mathbf{W}_L, \mathbf{W}_Q$ to construct a $\mathbf{W}^*$ which has small regularized loss. Recall the definition of the regularizers, after setting $r = n_k$:

$$\mathcal{R}_1(\mathbf{W}) = \mathcal{R}_1(\mathbf{W}; n_k) = \|f_L(\cdot; \mathbf{P}_{>k}\mathbf{W})\|_{L^2}^2 = \mathbb{E}_\mu\left[(f_L(\mathbf{x}; \mathbf{P}_{>k}\mathbf{W}))^2\right] \tag{24}$$

$$\mathcal{R}_2(\mathbf{W}) = \mathcal{R}_2(\mathbf{W}; n_k) = \|f_L(\cdot; \mathbf{P}_{\leq k}\mathbf{W})\|_{L^2}^2 = \mathbb{E}_\mu\left[(f_L(\mathbf{x}; \mathbf{P}_{\leq k}\mathbf{W}))^2\right] \tag{25}$$

$$\mathcal{R}_3(\mathbf{W}) = \mathcal{R}_3(\mathbf{W}; n_k) = \mathbb{E}_n\left[(f_L(\mathbf{x}; \mathbf{P}_{>k}\mathbf{W}))^2\right] \tag{26}$$

$$\mathcal{R}_4(\mathbf{W}) = \|\mathbf{W}\|_{2,4}^8 \tag{27}$$

Also, define the empirical loss of the quadratic model as:

$$\hat{L}^Q(\mathbf{W}) := \mathbb{E}_n[\ell(y, f_L(\mathbf{x}; \mathbf{W}) + f_Q(\mathbf{x}; \mathbf{W}))] \tag{28}$$

The following theorem is the central expressivity result:

**Theorem 2.** *For $\varepsilon_{min} > 0$, let $m \gtrsim \max(d^{3(k-1)}\varepsilon_{min}^{-4}, d^{10k})$, $n \gtrsim \max(d^k\varepsilon_{min}^{-2}, d^k \log d)$. With probability $1 - 1/poly(d)$, there exists $\mathbf{W}^*$ such that*

$$\hat{L}^Q(\mathbf{W}^*) \leq \varepsilon_{min} \tag{29}$$

*and:*

$$\mathcal{R}_1(\mathbf{W}^*) \lesssim m^{-\frac{1}{2}}d^{\frac{k-1}{2}} \qquad \mathcal{R}_2(\mathbf{W}^*) \lesssim 1 \qquad \mathcal{R}_3(\mathbf{W}^*) \lesssim m^{-\frac{1}{2}}d^{\frac{k-1}{2}} \qquad \mathcal{R}_4(\mathbf{W}^*) \lesssim d^{2(k-1)}. \tag{30}$$

The proof of this Theorem is presented in Appendix B.3, and again relies on the eigendecomposition of $\mathbf{\Sigma}$. As outlined in the introduction, we show $\mathbf{\Sigma} = \mathbf{Q}_1 \mathbf{\Lambda}_1 \mathbf{Q}_1^T + \mathbf{Q}_2 \mathbf{\Lambda}_2 \mathbf{Q}_2^T + \mathbf{Q}_3 \mathbf{\Lambda}_3 \mathbf{Q}_3^T$, where $\mathbf{\Lambda}_2$ are the medium eigenvalues and $\mathbf{\Lambda}_3$ are the small eigenvalues. Formally, $\mathbf{\Lambda}_2$ contain $\Theta(d^{-i+1})$ with multiplicity $\Theta(d^i)$, for integers $i \in [k+1, 2k]$, and the entires of $\mathbf{\Lambda}_3$ are equal to $1/m$ on average. This tells us that the medium directions $\mathbf{Q}_2$ are undesirable, as any $\Omega_m(1)$ movement in these directions will case $f_L(\mathbf{x}; \mathbf{W})$ to grow large. On the other hand, movement in a "sufficiently random" $\mathbf{Q}_3$ direction will minimally affect the population value of $f_L(\mathbf{x}; \mathbf{W})$.

To prove Theorem 2, we will construct $\mathbf{W}^*$ to be of the form $\mathbf{W}_L + \mathbf{W}_Q$, where $\mathbf{W}_L$ fits the low-degree term and $\mathbf{W}_Q$ fits the sparse term. The issue with this direct construction is that $\|\mathbf{W}_Q\|_F \gg 1$, so a priori $\mathbf{W}_Q$ can have a large effect on the linear term. We thus require $f_L(\mathbf{x}; \mathbf{W}_Q)$ to be small, both on the sample and over the population. The key insight is the following: since $\dim(\mathbf{V}_1) \ll \dim(\mathbf{V}_2)$, any sufficiently random direction lies almost entirely in $\mathbf{V}_2$. If $\mathbf{u}$ is this random direction, then $\mathbb{E}_n\left[(f_L(\mathbf{x}; \mathbf{u}))^2\right]$ is small. Similarly, the random direction $\mathbf{u}$ lies almost entirely in $\mathbf{Q}_3$. Since the $\mathbf{Q}_3$ directions have eigenvalues $1/m$ on average, $\mathbb{E}_\mu\left[(f_L(\mathbf{x}; \mathbf{u}))^2\right] \approx \frac{1}{m}\|\mathbf{u}\|^2 \ll 1$.

We thus consider a "sufficiently random" version of $\mathbf{W}_Q$ so that $\mathbf{W}_Q$ minimally affects $f_L$. Specifically, we consider the weight $\mathbf{SW}_Q$, where $\mathbf{S} \in \mathbb{R}^{m \times m}$ is a diagonal matrix of random signs. By definition $f_Q(\mathbf{x}; \mathbf{SW}_Q) = f_Q(\mathbf{x}; \mathbf{W}_Q)$, furthermore, we show $\mathbf{SW}_Q$ is now sufficiently random in that $\mathbb{E}_n\left[(f_L(\mathbf{x}; \mathbf{u}))^2\right], \mathbb{E}_\mu\left[(f_L(\mathbf{x}; \mathbf{u}))^2\right]$ are both small. This allows us to prove that in expectation over $\mathbf{S}$, a solution of the form $\mathbf{W}_L + \mathbf{SW}_Q$ acheives small regularized loss, which implies the existence of such a desirable solution via the probabilistic method.

## 4.2 Landscape

Define first and second-order stationary points [31] as follows:

**Definition 1.** $\mathbf{W}$ *is a $\nu$-first-order stationary point of $f$ if $\|\nabla f(\mathbf{W})\| \le \nu$.*

**Definition 2.** $\mathbf{W}$ *is a $(\nu, \gamma)$-second-order stationary point (SOSP) of $f$ if $\nabla^2 f(\mathbf{W}) \succeq -\gamma \mathbf{I}$.*

The following lemma is our central landscape result:

**Lemma 3.** *Let $r = d_k, \lambda_2 = \varepsilon_{min}, \lambda_3 = m^{\frac{1}{2}} d^{-\frac{k-1}{2}} \varepsilon_{min}, \lambda_4 = d^{-2(k-1)} \varepsilon_{min}$. Assume $m \ge n^4 d^{\frac{26(k+1)}{3}} \varepsilon_{min}^{-22/3}$, and $\nu \le m^{-\frac{1}{4}}$. Let $\mathbf{W}$ be a $\nu$-first order stationary point, and let $\mathbf{W}^*$ be the solution constructed in 2. Then,*

$$\mathbb{E}_\mathbf{S}\left[\nabla^2 L_\lambda(\mathbf{W})[\mathbf{SW}^*, \mathbf{SW}^*]\right] - \langle \nabla L_\lambda(\mathbf{W}), \mathbf{W} - 2\mathbf{W}_L^* + \mathbf{W}_L \rangle + 2L_\lambda(\mathbf{W}) - 2L_\lambda(\mathbf{W}^*) \lesssim \varepsilon_{min}. \tag{31}$$

As a corollary, we show that any $(\nu, \gamma)$-SOSP of $L_\lambda(\mathbf{W})$ has small loss.

**Corollary 1.** *Set $r, \lambda$ as in Lemma 3. Let $\hat{\mathbf{W}}$ be a $(\nu, \gamma)$-second-order stationary point of $L_\lambda(\mathbf{W})$, with $\nu \le m^{-1/2}, \gamma \le m^{-3/4}$. Then $L_\lambda(\hat{\mathbf{W}}) \lesssim \varepsilon_{min}$.*

The proofs of these results are deferred to Appendix C.

## 4.3 Optimization

We next invoke the main theorem of [30, 31], which is that perturbed gradient descent will find a $(\nu, \gamma)$-SOSP in $\text{poly}(1/\nu, 1/\gamma)$ time. The challenge with applying these results directly is that $L_\lambda$ is no longer smooth or Hessian-Lipschitz due to the $\mathcal{R}_4$ regularizer. To circumvent this, we first prove that the iterates in perturbed gradient descent are bounded in a Frobenius norm ball. Then, it suffices to use the bound on smoothness and Hessian-Lipschitzness in this ball to prove convergence. Our main optimization result, with proof in Appendix D, is the following:

**Theorem 3.** *There exists a choice of learning rate $\eta$ and perturbation radius $\sigma$ such that with probability $1 - 1/\text{poly}(d)$, perturbed gradient descent (c.f [31, Algorithm 1]) reaches a $(\nu, \gamma)$-SOSP within $\mathscr{T} = \text{poly}(m)$ timesteps.*

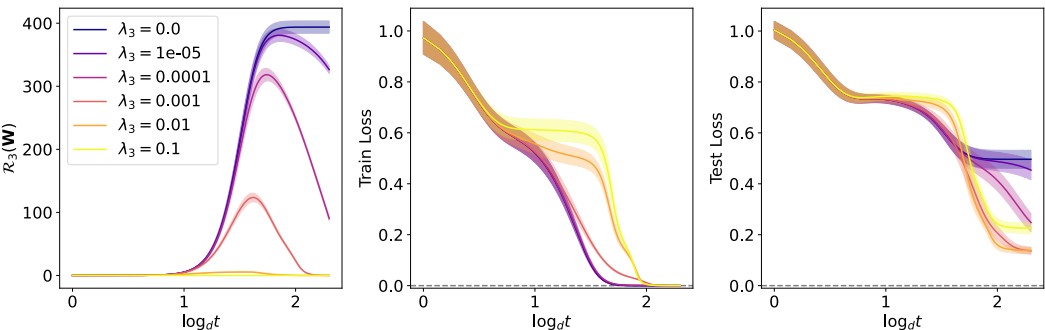

Figure 1: We train $f_L + f_Q$ with varying $\lambda_3$. When $\lambda_3$ is small, the NTK overfits the high degree signal and test error is large. When $\lambda_3$ is large, the QuadNTK can learn the high degree signal, and test error is smaller. Results are averaged over 5 trials, with one standard deviation shown.

### 4.4 Generalization

Finally, we conclude by showing that any $\hat{\mathbf{W}}$ with small $L_\lambda(\hat{\mathbf{W}})$ also has small test loss:

**Theorem 4** (Main generalization theorem). *Let* $r = d_k$, $\lambda_1 = m^{\frac{1}{2}} d^{-\frac{k-1}{2}} \varepsilon_{min}$, $\lambda_2 = \varepsilon_{min}$, $\lambda_3 = m^{\frac{1}{2}} d^{-\frac{k-1}{2}} \varepsilon_{min}$, $\lambda_4 = d^{-2(k-1)} \varepsilon_{min}$. *Assume* $m \gtrsim \varepsilon_{min}^{-4} d^{3(k-1)}$. *With probability* $1 - 1/poly(d)$ *over the draw of* $\mathcal{D}$, *any data dependent* $\hat{\mathbf{W}}$ *with* $L_\lambda(\hat{\mathbf{W}}) \leq C\varepsilon_{min}$ *has population loss*

$$L(\hat{\mathbf{W}}) \lesssim \varepsilon_{min} + \sqrt{\frac{d^k}{n}}. \tag{32}$$

The proof of this theorem is presented in Appendix E. The key is to use the small values of the regularizers at $\hat{\mathbf{W}}$ to bound the test loss. $\mathcal{R}_2$ and $\mathcal{R}_4$ are used to bound the Rademacher complexities of the linear and quadratic terms respectively, while $\mathcal{R}_1$ and $\mathcal{R}_3$ control the influence of the high degree component of the linear term on the population loss and empirical loss respectively.

## 5 Experiments

We conclude with experiments to support our main theorem. In Figure 1 we train the joint linear and quadratic model $f_L(\mathbf{x}; \mathbf{W}) + f_Q(\mathbf{x}; \mathbf{W})$ via gradient descent on the square loss, with signal $f^*(\mathbf{x}) = f_1(\mathbf{x}) + (\beta^T \mathbf{x})^2$ where $f_1(\mathbf{x}) = x_1 - 1$. Our covariates are dimension $d = 100$, we have $n = d^{1.5} = 1000$ samples, and our network has width $m = 10000$.

We train our model with the regularizer $\mathcal{R}_3(\mathbf{W})$ for varying values of $\lambda_3$. Rather than computing $\mathbf{\Sigma}$, we use the top $n_k$ right singular vectors of $\mathbf{\Phi}$ to estimate $\mathcal{R}_3$. We observe that for all $\lambda_3$ we reach near zero training error. However, when $\lambda_3$ is zero or very small, the model struggles to learn the entire signal, and test error plateaus near 0.6. In the leftmost pane, we plot the value of $\mathcal{R}_3(\mathbf{W})$ over the course of training. We observe that $\mathcal{R}_3(\mathbf{W})$ grows large for small $\lambda_3$, which implies that the model is using the "bad" directions in $\mathbf{V}_2$ to overfit to the training data. For large values of $\lambda_3$, however, $\mathcal{R}_3$ prevents $\mathbf{W}$ from moving in the "bad" directions to overfit the data. This is seen in the leftmost pane, where the value of $\mathcal{R}_3(\mathbf{W})$ is small. Instead, the parameter moves in the "good" $\mathbf{Q}_3$ directions, and the Quad-NTK term kicks in to fit the remaining component of the signal while generalizing. This leads to the consistently lower test loss (around 0.2) as shown in the rightmost pane.

**On the regularizer $\mathcal{R}_1$.** Our proof required the existence of $\mathcal{R}_1$ to prevent movement in the bad $\mathbf{Q}_2$ directions. While $\mathcal{R}_1$ being small is necessary for generalization, in Figure 1 we observe that we generalize well without explicitly regularizing $\mathcal{R}_1$. We hypothesize that the noise in the perturbed gradient descent update may be implicitly regularizing $\mathcal{R}_1$. A heuristic justification is as follows. Recall that any sufficiently random direction $\mathbf{u}$ lies mostly in $\mathbf{Q}_3$. Due to the rotational symmetry of the perturbations, one might expect that the solutions $\mathbf{P}_{<k}\hat{\mathbf{W}} + \mathbf{P}_k\hat{\mathbf{W}}\mathbf{S}$, over all choices of random signs $\mathbf{S}$, can be reached with roughly equal probability. Since this set of solutions is "sufficiently

random," in expectation over $\mathbf{S}$ it generalizes well, and thus gradient descent is likely to converge to a solution which generalizes well. In Appendix G we conduct experiments with both $\mathcal{R}_1$ and $\mathcal{R}_3$ to support the claim that explicit regularization of $\mathcal{R}_1$ is not needed to acheive low test error; however, proving this claim is left for future work.

**On the Low-Degree Plus Sparse Task.** Theorem 1 shows that two-layer neural networks trained via noisy gradient descent with a specific regularizer can more efficiently learn low-degree plus sparse target functions. In Figure 2 we show empirically that neural networks with standard initialization and trained via vanilla gradient descent efficiently learn a "dense quadratic plus sparse cubic." For varying values of $d$, we train a two-layer neural network to convergence and plot the smallest $n$ such that the test loss was $< 0.1$. We observe this minimal $n$ roughly scales with $d^2$, the optimal (in $d$) sample complexity. This provides evidence that standard networks trained with vanilla GD can effectively learn the low-degree plus sparse task, and hence that this is a sensible task to study and that our work presents a step towards understanding why standard neural networks perform better in practice more generally. See Appendix G for more details.

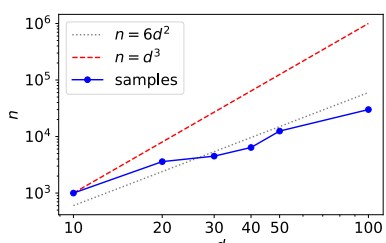

Figure 2: Neural networks optimally learn the "dense quadratic plus sparse cubic" task.

**On Second-Order Taylor Expansions of Standard Networks.** Throughout this paper we studied the quadratic Taylor expansion of a two-layer neural network. In Appendix G, we complement the results of [8] and show that for a standard architecture and data distribution, the quadratic Taylor expansion better approximates the optimization dynamics and acheives lower test loss than the linearization (NTK) does.

# 6 Discussion

The goal of this work is to better understand how neural networks can escape the NTK regime. By analyzing the eigendecomposition of the feature covariance matrix, we identified 3 sets of directions – ones that can fit low degree signal, "bad" directions which either cause the NTK to overfit or the test predictions to explode, and "good" directions in which the parameters can move a large distance. We then constructed a regularizer which encourages movement in these good directions, and showed how a network can jointly use the linear and quadratic terms in its Taylor expansion to fit a low-degree plus sparse signal. Altogether, we provided an end-to-end convergence and generalization guarantee with a provable sample complexity improvement over the NTK and QuadNTK.

As discussed above, one interesting direction of future work is to understand the role of $\mathcal{R}_1$. Other directions of future work include understanding whether our analysis can be used to leverage higher-order terms in the Taylor expansion, understanding the connection between the QuadNTK and feature learning, and investigating whether increasing the depth of the network can allow the NTK and QuadNTK to jointly learn a hierarchical representation [12].

# Acknowledgements

EN acknowledges support from a National Defense Science & Engineering Graduate Fellowship. JDL and EN acknowledge support of the ARO under MURI Award W911NF-11-1-0304, the Sloan Research Fellowship, NSF CCF 2002272, NSF IIS 2107304, NSF CIF 2212262, ONR Young Investigator Award, and NSF-CAREER under award #2144994. The authors would like to thank Alex Damian for helpful discussions throughout the project.

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
