# OpenReview forum: "Identifying good directions to escape the NTK regime and efficiently learn low-degree plus sparse polynomials "
_NeurIPS.cc/2022/Conference — NeurIPS 2022 Accept_

### Official Review · Reviewer_g4sF · 2022-06-27

**Rating:** 5
**Confidence:** 4
**Soundness:** 3 good
**Presentation:** 3 good
**Contribution:** 3 good

**Summary:**

This paper builds on a solid theory for the optimization and generalization of two-layer neural networks fitting polynomials. In particular, they jointly utilize the power of NTK and QuadNTK to fit target functions consisting of a dense low-degree term and a sparse high-degree term, demonstrating its sample complexity is superior to either NTK and QuadNTK.

**Questions:**

Please see the Weaknesses part.

**Limitations:**

Please see the Weaknesses part.

**Strengths And Weaknesses:**

**Strengths:**
* This paper improves upon the previous QuadNTK, showing that a special form of NN can avoid the kernel regime and achieve better sample complexity than any rotationally invariant kernels while fitting target functions consisting of a dense low-degree term and a sparse high-degree term. Compared to QuadNTK, it benefits from the linear term, providing its power to fit the dense polynomials.
* The paper is well-structured and provides a clear proof sketch.
**Weaknesses:**
* As mentioned by the author self, some assumptions, i.e., smoothness and nonzero Hermite/Gegenbauer coefficients of the activation function, boundedness of the loss function, sphere distribution of the inputs. These assumptions are mainly technical, but some of them seem to be crucial for the proof techniques, like nonzero Hermite/Gegenbauer coefficients, which might prevent the general impact and application of the proposed methods.
* The analysis of this paper heavily relies on the regularizers. Hence, my question is about the role of these regularizers. Are they just for the proof simplicity or necessary since we consider the quadratic approximation of the NN. If the answer is the latter, then it seems that there is something different from the model considered in this work might not reflect the practical success of general NNs.
* This paper provides great intuition about how to avoid the blowup of the training loss and population and the usage of the four regularizers. However, it would be great if the authors can provide the intuition why the sample complexity can be reduced from $\Omega(d^{k+1})$ to $O(d^k)$.
* This paper fixed the second layer parameters, which also limits the general impact of this work.
* (typo) The sign in line 273 is wrong.

---

> ### Author Response · Authors · 2022-08-02
> **Response to Reviewer g4sF**
>
> We would like to thank the reviewer for your detailed and thoughtful review. Please do let us know if you have any further questions or if anything is still unclear. We address specific comments below:
>
> > “As mentioned by the author self, some assumptions, i.e., smoothness and nonzero Hermite/Gegenbauer coefficients of the activation function, boundedness of the loss function, sphere distribution of the inputs.”
>
> These assumptions are common in previous work (see Ghorbani et al 2021, Montanari & Zhong 2020). The assumptions on the activation function are mild, and are satisfied by, for example, $\sigma(x) = \mathrm{sigmoid}(x - 0.5)$. The spherical data assumption is necessary to invoke the lower bound on the NTK. While generalizing this to arbitrary distributions is an important direction, it is still an open question to understand the eigenfunctions of the NTK for more general data distributions.
>
> > “Hence, my question is about the role of these regularizers. Are they just for the proof simplicity or necessary since we consider the quadratic approximation of the NN. If the answer is the latter, then it seems that there is something different from the model considered in this work might not reflect the practical success of general NNs.”
>
> We would like to re-emphasize the contribution of our work towards understanding the dynamics of GD in standard neural networks. An important open question is to understand how neural networks used in practice outperform their linearization. The previous empirical work: “Taylorized Training: Towards Better Approximation of Neural Network Training at Finite Width” (Bai et al 2020, https://arxiv.org/pdf/2002.04010.pdf) showed that in standard CNNs trained on CIFAR10, the second-order Taylor expansion of a neural network better tracks the GD dynamics than the linearization does. In Appendix G.2, we have added an additional experiment demonstrating that in a standard initialized CNN trained on the cats vs. horses task in CIFAR10, the second-order expansion better tracks GD dynamics. Furthermore, we demonstrate that both the linear term and quadratic term learn a nontrivial component of the signal.
>
> However, there do not exist any existing theoretical results which prove that both the linear term and quadratic term can be used to learn. While certain aspects of the problem setting (uniform-on-sphere data distribution, choice of regularization) indeed differ from the standard deep learning practice, our work is the **first result which identifies a setting in which the linear term and quadratic term can both provably learn a component of the signal**. This is an important step towards understanding how both terms can be leveraged to learn more generally.
>
> > “it would be great if the authors can provide the intuition why the sample complexity can be reduced from $\Omega(d^{k+1})$ to $O(d^k)$.
>
> The NTK is a kernel method, and thus cannot adapt to sparsity in the signal (think linear regression with l2 regularization). The reason why the QuadNTK can adapt to sparse signal is that the quadratic parameterization causes the QuadNTK Rademacher complexity bound to have a term which is $||\sum x_ix_i^T ||_{op}$. This operator norm, rather than the Frobenius norm which arises in the kernel Rademacher bound is very crucial. When the data is isotropic, the operator norm is a factor of $d$ smaller than the Frobenius norm, which leads to the improved sample complexity.
>
> > “This paper fixed the second layer parameters, which also limits the general impact of this work.”
>
> Previous works (Montanari & Zhong 2020, Bai & Lee 2020) have also fixed the second layer parameters. This is a reasonable assumption in such analyses, since there are $md$ parameters in the first layer and only $d$ parameters in the second layer. Therefore fixing the second layer does not significantly constrain the degrees of freedom of the model.
>
> We have also fixed the mentioned typo.

---

> > ### Comment · Reviewer_g4sF · 2022-08-07
> > **Thanks for the rebuttal**
> >
> > First of all, I am very grateful that the author addressed most of my questions. Below are some of my follow-up suggestions and minor questions:
> >
> > > The spherical data assumption is necessary to invoke the lower bound on the NTK.
> >
> > I think the spherical data assumption is also necessary for the upper bound provided by the author. The reason is actually mentioned by the author that to reduce the sample complexity, we should use the fact that the operator norm is smaller than the Frobenius norm by a factor $\sqrt{d}$. This is not true in general. At least we should assume that the data is nearly isometric (uniform in all directions).
> >
> > However, I like the intuition provided by the author that why linear + quadratic NTK can reduce the sample complexity. I believe this intuition holds for a set of general data distributions. It would be great if they could add this part in the revised version.
> >
> > > This is a reasonable assumption in such analyses, since there are $md$ parameters in the first layer and only $d$ parameters in the second layer.
> >
> > To me, this is not a good answer. It is possible that there is some significant movement in the last layer, which does not contradict the fact that the number of the parameters is much less than in the first layer. More importantly, this is indeed the case that there is a significant movement for the last layer (the last layer norm is an indicator in the NN training as observed in [1]).
> >
> > [1] Thilak, Vimal, et al. "The Slingshot Mechanism: An Empirical Study of Adaptive Optimizers and the Grokking Phenomenon." arXiv preprint arXiv:2206.04817 (2022).

---

> > > ### Author Response · Authors · 2022-08-08
> > > **Thank you for your response**
> > >
> > > Thank you very much for your response and for engaging with us on discussion of our paper.
> > >
> > > Below we address your remaining concerns:
> > >
> > > **Regarding spherical data assumption:**
> > >
> > > > At least we should assume that the data is nearly isometric (uniform in all directions)
> > >
> > > That is correct; the sample complexity improvement obtained by the QuadNTK requires the data to be isotropic, so that the operator norm can be smaller than the Frobenius norm. In our proof, the uniform-on-sphere assumption is required to use the exact characterization of the NTK, which we used to construct our regularizers. We will add discussion of this point to the revised version.
> > >
> > > **Regarding constraining the last layer weights**
> > >
> > > We remark that fixing the last layer weights and training the first layer is a standard assumption in a number of prior works. If both layers are trained, there is a slightly different form of the NTK and QuadNTK which comes from Taylor expanding both sets of parameters. We expect that our conclusions wouldn’t change in this setting, and fixing the last layer was chosen to simplify the analysis (a simplification which has been made in previous work)
> > >
> > > Please do let us know if you have any additional questions!

---

### Official Review · Reviewer_nemz · 2022-07-10

**Rating:** 6
**Confidence:** 4
**Soundness:** 3 good
**Presentation:** 4 excellent
**Contribution:** 2 fair

**Summary:**

The authors investigate very wide 2 layer neural networks and their learning dynamics on a special class of target distributions given by sparse polynomials. It has been established that neural networks in the NTK regime exhibit suboptimal sample complexities on such distributions, while QuadNTK, which incorporates a second order term, can improve said complexity. On the other hand, QuadNTK suffers from worse complexities on dense polynomials, compared to the NTK and is hence also not completely satisfactory. The authors show that combining NTK and QuadNTK together with specially designed regularizers can unite the best aspects of the two worlds, enabling the model to efficiently learn a combination of sparse and dense polynomials. The same cannot be achieved with neither NTK nor QuadNTK in isolation. The regularizers are specifically crafted such that weights only move within certain subspaces associated with big and small eigenvalues but ignoring mid-sized ones. While the large eigenspace allows to preserve the benefits from the NTK regime, the small eigenvalues enable large movement of neurons and thus feature learning.

**Questions:**

See section above.

**Limitations:**

As mentioned above, the relevancy of these results beyond the setting of polynomial regression is not clear and not addressed in the main text.

**Strengths And Weaknesses:**

**Strengths:**
1. The paper is very well-written and nicely structured, making it easy to read.
2. The differences between the lazy learning regime and the “actual” regime in which neural networks usually operate in practice are still not well-understood and making progress in this area would be very helpful to the community. The paper definitely takes steps towards this direction and identifies benefits of operating outside of the kernel regime.
3. The idea of dividing the spectrum into three categories, the large part where the NTK reduces the test error, the middle part responsible for overfitting and the small part where the quadratic term reduces the test error, is very appealing and intuitively makes sense.

**Weaknesses:**
1. Only investigating a polynomial regression setting seems like a very particular restriction and I wonder how much of these theoretical findings translate to real-world problems such as image classification. Is it a good idea (empirically speaking) to ignore the mid-sized eigenvalues for practical tasks (does such a clear clustering of eigenvalues even emerge for non-uniform inputs over the sphere?) or do those regularizers only make sense in the context of polynomial regression? The authors only evaluate their method on polynomial regression tasks but no other benchmark such as CIFAR10 or MNIST. Again, for these distributions the formulation might not even make sense but this would hence pose a significant limitation. I would appreciate if the authors could comment on this.
2. Finding particular distributions where the NTK provably performs worse than standard neural networks is a very sensible thing to do. To my understanding, the polynomial regression task has been already identified as being suboptimally learnt by NTK while the QuadNTK can improve over it in sparse settings. This makes the contribution of this work less clear to me. The fact that QuadNTK is suboptimal for dense polynomials is interesting but I am not sure what is the significance of developping another (somewhat artificially-designed) approach that fixes this particular issue, other than enabling better polynomial regression. Especially if point 1. holds and the approach doesn’t easily extend to more realistic setups with empirical improvements, I am not sure what we can learn from these results.
3. This brings me to my next point/question. The goal (as stated in the introduction) is to understand why NTK might perform suboptimal compared to realistic, finite-width networks. In this work (and the QuadNTK work), it is shown that higher order approximations, together with special regularizers and specific initialization schemes can outperform the NTK in terms of sample complexity. This however somehow does not answer the original question yet, do these new approaches also mimic better what is actually happening in finite networks? This is not entirely obvious to me, given the somewhat unintuitive form of the regularizers, exotic initialization schemes, perturbed SGD instead of SGD etc. For instance, if you were to train finite-width networks on these polynomial tasks, would they be able to efficiently learn both sparse and dense polynomials? Are there similar, bad directions in parameter space where we should not move? Would your regularisers also make sense and lead to performance boosts in this setting? As far as I understood, experiments only concern the newly developed approaches yet no standard setup. Experiments with standard neural networks would really complete the picture in my opinion.

---

> ### Author Response · Authors · 2022-08-02
> **Response to Reviewer nemz (1/2)**
>
> We would like to thank the reviewer for your detailed and thoughtful review. We hope our responses below address your concerns and that you would consider raising your score. Please do let us know if you have any further questions or if anything is still unclear.
>
> First, we would like to re-emphasize the contribution of our work towards understanding the dynamics of GD in standard neural networks. An important open question is to understand how neural networks used in practice outperform their linearization. The previous empirical work “Taylorized Training: Towards Better Approximation of Neural Network Training at Finite Width” (Bai et al 2020, https://arxiv.org/pdf/2002.04010.pdf) showed that in standard CNNs trained on CIFAR10, the second-order Taylor expansion of a neural network better tracks the GD dynamics than the linearization does. In Appendix G.2, we have added an additional experiment demonstrating that in a standard initialized CNN trained on the cats vs. horses task in CIFAR10, the second-order expansion better tracks the full network's GD dynamics. Furthermore, we demonstrate that both the linear term and quadratic term learn a nontrivial component of the signal.
>
> However, there are no existing theoretical results which prove that both the linear term and quadratic term can be used to learn a component of the signal. While certain aspects of the problem setting (uniform-on-sphere data distribution, choice of regularization) indeed differ from the standard deep learning practice, our work is the **first result which identifies a setting in which the linear term and quadratic term can both provably learn a component of the signal**. This is an important step towards understanding how both terms can be leveraged to learn more generally. We additionally comment that both the uniform-on-sphere data assumption and two layer network with symmetric initialization are standard in the deep learning theory literature.
>
> Next, we address some of the reviewer’s specific concerns:
>
> > “Only investigating a polynomial regression setting seems like a very particular restriction and I wonder how much of these theoretical findings translate to real-world problems such as image classification….The authors only evaluate their method on polynomial regression tasks but no other benchmark such as CIFAR10 or MNIST.”
>
> Firstly, we would like to point out that polynomial regression has been a sensible benchmark for investigating deep learning theory, and a number of previous works have used this task to either understand the performance of the NTK or show improvements over the kernel regime. For examples, see the set of papers (Arora et al 2019, Bai & Lee 2020, Ghorbani et al 2019, 2020, Hu et al 2020, Wei et al 2018). Going beyond polynomial regression is a major open question in the field and is not a focus of our work.
>
> While real-world target tasks such as CIFAR classification may not be expressed as polynomials of the input, many sample complexity comparisons between different models of training do translate to real world data distributions. For example, while the QuadNTK paper (Bai & Lee, 2020) shows that the QuadNTK can outperform the NTK on learning sparse polynomials, the later empirical work (Bai et al, 2020) shows that the linear+quadratic model outperforms the NTK on CIFAR as well.
>
> We also have a strong theoretical reason for focusing on the polynomial learning task. For uniform-on-sphere data, the eigenfunctions of the NTK are the Gegenbauer polynomials; this fact is used in Ghorbani et al (2021) to prove lower bounds on the learning capabilities of the NTK, and we leverage it to construct our regularizers. For other data distributions (i.e the “natural” distribution of CIFAR images), it is an open question to understand what the eigenfunctions of the NTK even are. Thus from a theory perspective, it is only possible to give provable guarantees for the polynomial learning task. However, there is reason to believe that polynomials are the “correct” basis of functions for arbitrary data distributions. For example, Hu et al (2020) (https://papers.nips.cc/paper/2020/file/c6dfc6b7c601ac2978357b7a81e2d7ae-Paper.pdf) shows that for a more general class of distributions the top eigenfunctions of the NTK are linear functions (degree 1 polynomials) of the input, and are thus learned in the early stages of training.
>
> (continued in next comment:)

---

> > ### Author Response · Authors · 2022-08-02
> > **Response to Reviewer nemz (2/2)**
> >
> > >“The fact that QuadNTK is suboptimal for dense polynomials is interesting but I am not sure what is the significance of developping another (somewhat artificially-designed) approach that fixes this particular issue, other than enabling better polynomial regression.”
> >
> > We would like to emphasize the 2 key improvements of our work over the original QuadNTK work.
> > 1. As you mentioned, the QuadNTK cannot learn arbitrary dense polynomials, and thus our work presents an method where neural networks can provably learn a larger class of functions.
> > 2. Perhaps more importantly is that the original QuadNTK paper used a randomization trick to delete the linear term, so that the network could be approximated by **only the quadratic term**. In practice no such randomization trick is used, and the Taylorization of a real neural network contains both a linear term and quadratic term. Our paper is the first work which utilizes **both** terms to learn. Since standard neural networks also use both terms to learn, we believe that our work makes progress towards understanding the dynamics of GD more generally.
> >
> > >“...do these new approaches also mimic better what is actually happening in finite networks? This is not entirely obvious to me, given the somewhat unintuitive form of the regularizers, exotic initialization schemes, perturbed SGD instead of SGD etc.”
> >
> > The property of finite networks we aim to explain is the ability to use both the linear term and quadratic term to learn a component of the signal. As discussed above, our work is the first which provably guarantees both the linear and quadratic terms learn a component of the signal.
> >
> > >”if you were to train finite-width networks on these polynomial tasks…”
> >
> > We remark that our theory requires width $m$ to be polynomial in the input dimension $d$. In the NTK literature, the width required has gradually decreased from a large polynomial in $d$ in Du et al (2018) (https://arxiv.org/pdf/1810.02054.pdf) to the minimal overparameterization in Montanari & Zhong (2020) (https://arxiv.org/pdf/2007.12826.pdf). We believe similar techniques can be used to improve the dependence on width in our setting.
> >
> > >”...experiments only concern the newly developed approaches yet no standard setup”
> >
> > This is a theoretical paper, and thus the simulations in section 5 are designed to verify Theorem 1, by matching the setup of the theorem as closely as possible. We remark that all of the related works which provide convergence and generalization guarantees for two-layer neural networks beyond the NTK regime make some modification to the standard learning paradigm (i.e input distribution, non-standard network architecture, choice of regularization), the reason being that developing an overall convergence theory beyond the NTK regime still remains an open problem. In addition to (Bai & Lee 2020, Chen et al 2020) using randomization to isolate the QuadNTK, other examples include: Du & Lee (2018) and Ghorbani et al (2019) using quadratic activations, Li et al (2020) using truncated GD, Daniely et al (2020) considering a biased input distribution, and Malach et al (2021) considering unconventional “threshold-based” networks. We further remark that it is not the standard practice in this line of work to verify such algorithmic choices on standard architectures or datasets such as CIFAR10.

---

> > > ### Comment · Reviewer_nemz · 2022-08-05
> > > **Response to Author's Rebuttal**
> > >
> > > Thank you very much for the answer, it helped me understand the relevance of this work!
> > >
> > > **Agreeing**
> > > 1) I agree that showing how leveraging both the linear and quadratic term is beneficial for training and generalization is very important, I somehow missed this in the original text (although it was stated), thanks for the clarification. I agree that the artificial randomisation trick in the QuadNTK work to get rid of the linearisation term is very unsatisfactory and the setup used in this paper resembles practice more.
> > >
> > > 2) While I still believe that polynomial regression is a bit unrealistic, in light of point 1) this weakness seems less relevant to me now.
> > >
> > > 3) Thank you for adding the experiment in the appendix, it also convinces me that both linear and quadratic term are relevant in neural network training in practice. Test loss is a bit difficult to interpret, did you also compute the corresponding test accuracies?
> > >
> > > **Disagreeing**
> > > 1) I would still like to understand how "realistic" finite-width neural networks perform on the task of polynomial regression. In my opinion, testing whether standard neural networks can learn both dense and sparse polynomials is a crucial experiment since the goal of this paper is to mimic their behaviour better by using both the linear and the quadratic term. If standard networks cannot learn both settings, then I don't see the point of trying to fix that in the large-width regime.
> > > Let me know if I'm missing something!
> > >
> > > If you can clarify my remaining doubt I would be very happy to raise my score!

---

> > > > ### Author Response · Authors · 2022-08-08
> > > > **Thank you for your response**
> > > >
> > > > Thank you very much for your response and for engaging with us on discussion of our paper.
> > > >
> > > > Regarding your additional comment:
> > > >
> > > > >I would still like to understand how "realistic" finite-width neural networks perform on the task of polynomial regression.
> > > >
> > > > This is indeed a valid point. To demonstrate that finite-width networks can indeed learn such functions, we trained a 2-layer neural network on the task of fitting a dense quadratic + a rank 1 cubic. Our setup is “realistic” in that we use the standard pytorch initialization, a width of 100, and train via vanilla GD with learning rate 0.05 (no regularization or other differences to the algorithm). For varying values of dimension from 10 to 100, we calculate the minimum number of samples $n$ needed to obtain a test loss of <0.1. We added a log-log plot of $n$ versus dimension $d$ in Appendix G.3, and observe that $O(d^2)$ samples are needed to learn. The NTK lower bound is $\Omega(d^3)$, while the minimax complexity in $\Theta(d^2)$, so this provides stronger evidence that “realistic” finite-width networks can learn these “low-degree dense+high-degree sparse” functions with the optimal sample complexity.
> > > >
> > > > To clarify test loss: The empirical setup in Appendix G.2 uses square loss, with cats having label 1 and horses having label -1. Therefore the null predictor achieves a test loss of 1.0 and test accuracy of 50%, while the optimal predictor has a test loss of 0.0 and test accuracy of 100%. We have now added test accuracies to Table 1.
> > > >
> > > > Please do let us know if you have any additional questions!

---

> > > > > ### Comment · Reviewer_nemz · 2022-08-09
> > > > > **Response**
> > > > >
> > > > > 1) Thank you very much for performing this additional experiment. This makes perfect sense to me now and nicely illustrates that the (at first sight artificial) task of studying sparse + dense polynomial regression is very interesting since QuadNTK and NTK both cannot learn it while finite NNs can. This convinces me that the combination of linear + quadratic is indeed important here and completes the picture.
> > > > >
> > > > > 2) Thanks for adding the accuracy values. One final question: What accuracy does the quadratic term $f_Q$ reach in isolation?
> > > > >
> > > > > I have raised my score from 4 to 6.

---

### Official Review · Reviewer_t53d · 2022-07-11

**Rating:** 7
**Confidence:** 3
**Soundness:** 4 excellent
**Presentation:** 4 excellent
**Contribution:** 3 good

**Summary:**

The paper focuses on escaping the NTK regime for gradient descent and provided a comprehensive theoretic analysis toward understanding how a two-layer neural network could escape the NTK regime. Via eigendecomposition of the covariance matrix, the parameter space is partitioned into large, medium (bad) and small (good) directions. With the theoretical guide toward “good” directions, the authors develop a regularizer to encourage parameters moving in these directions. They also demonstrate that the linear and quadratic terms of the Taylor expansion can be used together to fit target functions and leverages from both NTK and QuadNTK to achieve provable improved sample complexity.

**Questions:**

See weakness and minor issues above.

**Limitations:**

The authors discussed the limitations of their work with regard to the assumptions made for their main theorem and the discussion is intact and reasonable to me.

**Strengths And Weaknesses:**

Strengths:
1. The paper is well motivated and presented, with detailed explanations of the current limitations of NTK and QuadNTK and clear illustrations of why the focus of this paper raises interest. The structure and logic of the paper are easy to follow and the proof sketch is coherent and helpful for understanding. Generally, the writing of the paper is excellent.
2. The contributions of this paper are well supported by solid evidence both theoretically and empirically. The theoretical proofs are solid as far as I read into, and the codes for the empirical results are available and runnable.

Weakness:
1. The introduced regularizer also introduces many hyperparameters $(\lambda_1,\lambda_2,\lambda_3,\lambda_4)$, which are not fully discussed in the paper. Does this bring limitations to the regularizer as there are many hyperparameters to attend to? In the experiments, there are ablation studies done on varying values of $\lambda_3$ and in the Appendix there are ablation studies on $\lambda_1$. What about the other hyperparameters $(\lambda_2,\lambda_4)$?

Minor Issues:
1. The legend for figure 1 is a bit unclear since the paper earlier defined $\lambda = (\lambda_1,\lambda_2,\lambda_3,\lambda_4)$, but the experiment is on varying values of $\lambda_3$. It could be clearer by changing $\lambda$ to $\lambda_3$ in the figure. Similarly in Appendix G figure 2, $\lambda$ -> $\lambda_1$.
2. The following work is closely related to the interest of this paper on NTK analysis. The authors may find it valuable to be included in the discussion of related works: Chen et al. "A generalized neural network tangent kernel analysis for two-layer neural networks." Advances in Neural Information Processing Systems 33 (2020).

---

> ### Author Response · Authors · 2022-08-02
> **Response to Reviewer t53d**
>
> We would like to thank the reviewer for your detailed and thoughtful review. Please let us know if you have any further questions or if anything is still unclear. We address specific comments below:
>
> > “What about the other hyperparameters $(\lambda_2, \lambda_4)$?”
>
> $\lambda_2$ and $\lambda_4$ act similarly to weight decay terms, and are only necessary for the proofs. The experiments in Figure 1 and 2 set $\lambda_2, \lambda_4$ to 0. The reason we need these regularizers in the proof is that we rely on the second-order stationary point convergence argument in Jin et al (2017, 2021), where an important step is to localize the minimizer (Lemma 28). Additionally, the terms are necessary for the Rademacher complexity argument. In these non-convex optimization settings, it is quite difficult to prove convergence without such regularization terms.
>
> We have updated our paper to fix the typo in the Figures and to add the Chen et al (2020) reference to the related work.

---

> > ### Comment · Reviewer_t53d · 2022-08-05
> > **Thanks for the rebuttal**
> >
> > Thanks for the explanation on the regularizers. It clarified a lot of my confusion. Personally, I believe that the paper is very interesting, informative and well-presented and thus increased my score from 6 to 7.

---

### Author Response · Authors · 2022-08-09
**Revision Uploaded**

We’re grateful to all the reviewers for their comments throughout this discussion period. We’ve uploaded a revised version of our paper with the following changes from the original submission:
- Fixed the various typos pointed out by the reviewers.
- Added Appendix G.2, which contains experiments on CIFAR10 showing that CNNs use both the linear and quadratic term to learn signal.
- Added Appendix G.3, which shows that finite-width two-layer neural networks can learn dense quadratic plus sparse cubic functions with optimal sample complexity.
- We’ve also attached the code for the additional experiments.

---

### Meta-Review · Area_Chair_irpH · 2022-08-27

**Recommendation:** Accept
**Confidence:** Certain

**Metareview:**

This paper studies the learning dynamics of two-layer neural networks beyond the NTK regime for learning low-degree plus sparse polynomials. The author response and discussion have addressed most of the reviewers’ questions and concerns. While some reviewers think the polynomial regression setting is a bit limited, all the reviewers agree that the results are interesting and significant.  Therefore, I recommend acceptance.

**Award:**

No

---

### Decision · Program_Chairs · 2022-09-14

Accept